# NEURAL NETWORKS DECODED: TARGETED AND ROBUST ANALYSIS OF NEURAL NETWORK DECISIONS VIA CAUSAL EXPLANATIONS AND REASONING

## ABSTRACT

Despite their continued success and widespread adoption, deep neural networks (DNNs) remain enigmatic "black boxes" due to their complex architectures and opaque decision-making processes taking place within, which poses significant trust challenges to critical applications. While various methods have been proposed to address this lack of interpretability, existing solutions often offer inconsistent, or overly simplified explanations, or require alterations that compromise model performance. In this work, we introduce TRACER, a novel explainability method grounded in causal inference theory and designed to shed light on the causal dynamics underpinning DNN decisions without altering their architecture or compromising their performance. We further propose an efficient methodology for counterfactual generation, offering contrastive explanations for misclassifications, thereby identifying potential model biases. Through comprehensive evaluations across diverse datasets, we demonstrate the superiority of TRACER compared to prevalent explainability methods, and underscore its ability to transcend explainability from mere associations to causal relationships. We subsequently highlight TRACER's potential to enable the creation of highly compressed and highly efficient models, showcasing its versatility in both understanding and optimizing DNNs.

## 1 INTRODUCTION

Deep neural networks (DNNs) have demonstrated transformative potential across various applications, notably image classification (Krizhevsky et al., 2012), medical diagnostics (Esteva et al., 2017), and complex pattern recognition (LeCun et al., 2015), even surpassing human expertise in certain domains (He et al., 2016; Silver et al., 2016; Rajpurkar et al., 2017). Yet, their inherent complexity conceals the underlying mechanisms behind their decision-making, rendering them as "black boxes" that present transparency and trust concerns, impeding their adoption in sectors requiring clear reasoning, such as healthcare and cybersecurity (Rudin, 2019; Castelvecchi, 2016; Doshi-Velez & Kim, 2017; Zeiler & Fergus, 2014; Ribeiro et al., 2016; Papernot & McDaniel, 2018; Goodfellow et al., 2014; Zhang et al., 2021; Lipton, 2018). Neural Network Explainability, pivotal in Explainable Artificial Intelligence (XAI), aims to clarify DNN decision-making, and thereby to ensure trust, ethical application, and bias mitigation. Although strategies like saliency maps (Simonyan et al., 2013; Zhou et al., 2015), Grad-CAM (Selvaraju et al., 2017), LIME (Ribeiro et al., 2016), and SHAP (Lundberg & Lee, 2017) have been developed towards this end, they often suffer from inconsistencies, over-simplification, or architectural constraints, with many overlooking specific prediction intricacies. All this underscores an ongoing challenge in model understanding (Bach et al., 2015; Baehrens et al., 2010; Ba & Caruana, 2014; Rudin, 2019).

In this paper we introduce TRACER, a new technique centered around causal inference theory (Pearl, 2009), which sheds light on how AI systems process inputs to derive decisions. Recognizing that conventional accuracy metrics based solely on validation datasets may not be indicative of a model's performance in real-world settings and drawing inspiration from causal hierarchy, our approach brings about a paradigm shift in neural network explainability by expanding our understanding of not just what happened, but why it happened, and what could have happened under different conditions. In essence, this frames the explainability of neural networks as a causal discovery and counterfactual inference problem, where for any given sample and a target neural network, we observe

all intermediate and final outputs during the inference process for the sample, its generated set of interventions, and its counterfactuals. By aggregating multiple such instances, TRACER provides interpretability to state-of-the-art models without necessitating any re-training or modification to their architectures, thus preserving their performance. We further propose an efficient approach for counterfactuals generation, which can be used to provide contrastive explanations for misclassified samples, enabling the identification of potential model blind spots and biases, thereby addressing the overarching issue of trustworthiness.

We evaluate TRACER on the MNIST dataset (Deng, 2012), presenting explanations for correct and misclassified samples, as well as a counterfactual analysis for misclassifications. We further perform comprehensive evaluations on image and tabular datasets, contrasting TRACER's performance against that of prevalent explanation techniques. Finally, we highlight our approach's potential for global explainability, demonstrating its ability to uncover redundancies in neural architectures and to aid in the creation of optimized, compressed models.

## 2 RELATED WORK

Various techniques have been developed for DNN interpretability, typically categorized by explainability scope, implementation stage, input/problem types, or output format (Angelov et al., 2021; Adadi & Berrada, 2018; Vilone & Longo, 2021). In pursuit of DNN transparency, early endeavours like saliency maps by Zhou et al. (2015) visually highlighted key features in input data. However, these visual explanation methods, including Grad-CAM (Selvaraju et al., 2017) and Layer-wise Relevance Propagation (LRP) (Bach et al., 2015), often produce inconsistent or coarse explanations, or require structural model changes, sometimes compromising performance or overlooking individual nuances crucial for true comprehension (Rudin, 2019). Model-agnostic approaches, such as LIME (Ribeiro et al., 2016) and SHAP (Lundberg & Lee, 2017) offer explanations by approximating model decision boundaries. However, these methods also face challenges such as resource intensiveness or inconsistencies in local explanations. While some research attempts to simplify complex DNNs to improve their interpretability (Ribeiro et al., 2016; Frosst & Hinton, 2017; Che et al., 2016), these efforts often introduce a compromise on performance, as the simpler models do not always capture the nuances of their complex counterparts.

Different from traditional methods that emphasize associations or correlations, causal inference techniques probe deeper, seeking to understand not just statistical correlations but uncover the true cause-effect relationships between variables. The idea of merging causal inference with AI is an emerging perspective, advocating for a more robust form of explainability. Rooted in the foundational work by Pearl (2009), prior works on causal inference for AI have primarily revolved around the use of causal diagrams and structural equation models to gain such associative understanding (Pearl, 2009; Xia et al., 2021; Kenny et al., 2021; Chou et al., 2022; Geiger et al., 2022; Yang et al., 2019).

In contrast to the aforementioned methods, rather than merely highlighting influential features or approximating decision boundaries, our TRACER approach seeks to unravel the causal dynamics that steer DNN decisions, without the need for altering the model or compromising its performance. Furthermore, by introducing counterfactual explanations, we illuminate not only why specific decisions occurred but also expound potential decision outcomes under altered circumstances. This leap from feature importance to causal relationships sets TRACER apart in the landscape of NN explainability.

## 3 METHODOLOGY

DNNs employ layers of interconnected neurons to approximate intricate and multifaceted functions, thus achieving their notable prowess in various tasks. However, to discern the intricacies of such architectures, we must consider not only the individual computations, but also causal relationships embedded throughout the network. With TRACER, we aim to expose such mechanisms, focusing on the causal dynamics that steer the network's decisions. To this end, we structure our methodology around two primary axes: (1) causal discovery, where we analyze the interactions and dependencies within the DNN to map out the causal pathways, and (2) counterfactual generation, where we simulate alternative scenarios to identify potential biases or blind spots of the models. Next, we delve into the specifics of these axes and establish a bridge between raw model outputs and rich, causal explanations,

thereby ensuring that users and practitioners can not only trust but also understand and refine a neural network's decision-making processes. Figure 1 gives a high-level overview of our proposed approach.

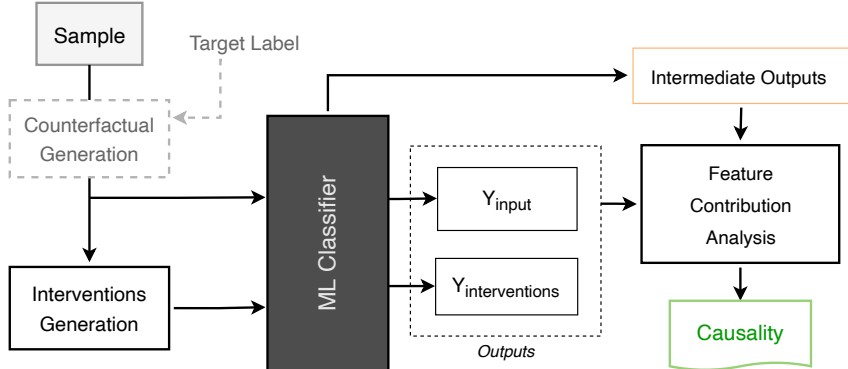

Figure 1: Overview of TRACER. Interventions on input samples (or their counterfactuals) are used to determine the effects of individual features on the intermediate and final outputs of the model. From this analysis, the causal mechanisms underpinning the decision-making processes are uncovered.

## 3.1 CAUSAL THEORY

Causal theory provides the means to quantify and dissect cause-effect relationships, offering a departure from mere observational statistics to tackle questions about interventions and counterfactuals. The language of Structural Causal Models (SCMs) offers a robust mathematical representation to formalize these relationships. Specifically, an SCM is characterized by:

- A set $U$ of exogenous variables that account for external influences not explicitly modeled. In DNN settings, we consider exogenous variables to exist for layers where randomness or other external factors are introduced.

- A set $V$ of endogenous variables that are determined by variables in the model.

- A set $F$ of functions $\{f_1, f_2, \ldots, f_n\}$ such that every $V_i \in V$ is associated with a function $f_i$ that determines its value based on a particular subset of $U \cup V$.

Mathematically, this can be expressed for each $V_i$ as: $V_i = f_i(pa(V_i), U_i)$, where $pa(V_i)$ represents the parents of $V_i$ in the associated causal graph, and $U_i$ is a subset of $U$.

Building on the foundation of SCMs, Pearl's Causal Hierarchy further refines our understanding by classifying causal knowledge into three distinct levels, namely: **Association (Observational)**, **Intervention (Action)**, and **Counterfactual (Retrospective)**. Applying this to the TRACER framework:

- **Association**: We extract dependency structures from the DNN activations and layers, formulating a mathematical representation, $P(Output \mid Layer_i)$, where $Layer_i$ represents specific layers in the DNN.

- **Intervention**: By selectively manipulating feature values, we can estimate the intervention distributions $P(Output \mid do(feature_j = value))$ to understand the effect of particular features on the final decision.

- **Counterfactual**: We can explore alternative (or hypothetical) input scenarios and compute the counterfactual distribution, $P(Output_y \mid do(Input = x), \ Input = x')$, which quantifies the model's output distribution if a certain input was set to a particular value, given that we actually observed another input.

Through the lens of causal reasoning, TRACER provides a grounded and rigorous methodology for untangling the intricate causal dynamics within deep neural networks.

## 3.2 CAUSAL DISCOVERY

To discover a faithful representation of the causal mechanisms underpinning DNN models, given an input and a target classifier, we perform an intervention-based analysis, where we actively change the values of the classifier's internal representations in systematic ways and study the effects. Such interventions on the input's representations allow us to measure the effects of specific changes on the output representation (Feder et al., 2021; Pryzant et al., 2020). By observing the internal states and outputs of the classifier, we can deduce how specific components contribute to the final decision-making process. This not only offers an understanding of the model's causal structure but also identifies key nodes or connections that highly influence the model's predictions. Furthermore, by collecting the observed effects of all interventions, we can establish a comprehensive causal map. This map visualizes the interplay of different network components and their collective influence on the classification outcomes. These detailed visualizations can potentially be instrumental for debugging or refining classifiers, or even for designing more interpretable neural network architectures.

### 3.2.1 INTERVENTIONS

In our DNN analysis, interventions are crucial for isolating and understanding the causal significance of specific input features or patches. Given an input vector $x \in \mathbb{R}^d$, where $d$ denotes the dimensionality of the input space, an intervention is simulated by replacing a subset of the components of $x$ with a predetermined baseline value $b$. For a specified subset of indices $I \subseteq \{1, \ldots, d\}$ corresponding to the features or patches under intervention, the intervened input $x'$ is formulated as:

$$\text{for } i = 1, \ldots, d, \quad x'_i = \begin{cases} b, & \text{if } i \in I; \\ x_i, & \text{otherwise.} \end{cases}$$

By performing such interventions, we effectively nullify or modify the influence of the selected features or patches, allowing for the evaluation of their causal effect on the model's output. Through these controlled perturbations, we can discern which features are causally pivotal for the model's decisions, and measure the depth of their influence.

These baseline values carry significant importance in our intervention framework. Much like in cooperative game theory where Shapley values (Shapley et al., 1953) use a baseline to understand the contribution of each player by averaging their marginal contributions across all possible coalitions, our baseline operates as a neutral point of reference. Specifically, in our interventions, the baseline value serves to counteract or neutralize the impacts of the specific features being altered. This allows us to isolate the original input's influence on the output without the bias introduced by those features. By contrasting the results from such intervened input with the original, we gain deeper insights into the causal relationships between input features and model outputs.

### 3.2.2 SIMILARITY BETWEEN DNN REPRESENTATIONS

For all samples processed by the classifier, the collected intermediate and final outputs are used to compare representations between different layers. One prevalent approach for measuring the similarity between high-dimensional representations is Centered Kernel Alignment (CKA).

CKA is a metric that quantifies the similarity between two sets of features by computing the alignment between their respective kernel matrices. Given feature matrices $X$ and $Y$, their kernel matrices, denoted by $K_X$ and $K_Y$, respectively, are aligned and quantified using the Hilbert-Schmidt Independence Criterion (HSIC), which measures the dependence between two sets of variables.

$$\text{CKA}(X, Y) = \frac{\text{HSIC}(X, Y)}{\sqrt{\text{HSIC}(X, X) \times \text{HSIC}(Y, Y)}},$$

$$\text{where} \quad \text{HSIC}(X, Y) = \frac{1}{(n-1)^2} \text{Tr}(K_X H K_Y H).$$

Here, $H$ is a centering matrix given by $H = I - \frac{1}{n}\mathbf{1}\mathbf{1}^T$, with $n$ being the number of samples, $I$ the identity matrix, and $\mathbf{1}$ a vector of ones. $\text{Tr}(\cdot)$ denotes the trace of a matrix.

There are several advantages to using CKA for representation similarity:

- **Normalization:** CKA provides a normalized measure, ensuring the similarity scores lie in the 0–1 range, where 0 indicates complete dissimilarity and 1 indicates identical representations. This normalization facilitates direct comparisons across different model layers.

- **Non-linear Relationships:** Unlike similarity metrics that focus solely on linear correlations, CKA captures both linear and non-linear relationships, a crucial property when analyzing deep neural networks known for their intricate non-linear transformations.

- **Robustness:** The utilization of kernels allows CKA to operate in a richer feature space, providing a more comprehensive similarity measure.

Given these properties, particularly the ability to capture non-linear relationships inherent between feature representations, CKA stands out as the most suitable choice for our similarity analysis of DNN representations. Upon obtaining the similarity measures, the CKA matrix undergoes filtering to identify layers exhibiting high mutual similarity. To derive causalilty from the similarity measures, we proceed by grouping layers based on their filtered CKA matrices. To this end, we construct a binary (merged) CKA matrix wherein a value of 1 is assigned if the filtered CKA value is non-zero, and 0 otherwise. Formally:

$$\text{Merged CKA}(X, Y) = \begin{cases} 1, & \text{if } \text{CKA}(X, Y) \geq 1 - \epsilon; \\ 0, & \text{otherwise}, \end{cases}$$

where $\epsilon$ represents a predetermined threshold, defining the maximum acceptable dissimilarity for two layers to be considered alike. In the context of our causal analysis, such similarity suggests that these layers contribute to a shared causal node.

Based on this merged matrix, we introduce the notion of "Layer Groups". A Layer Group is established by conjoining layers that are both adjacent and exhibit high mutual similarity, which is discerned from their respective values being set to 1 in the merged CKA matrix. This process encapsulates the layers into cohesive groups, where each group represents a distinct causal node in the underlying causal mechanism of the network. Furthering our causal analysis, we establish the existence of causal links between any two causal nodes if either the two corresponding layer groups are adjacent, or if the layers within them exhibit strong similarities, as identified by their filtered or merged CKA values. By adopting these criteria for causality, we not only capture the immediate dependencies inherent to the network's architecture, but also uncover the deeper relationships arising from its representation similarities. This enriched perspective allows for a deeper understanding of the interplay between layers, and how they collectively shape the decision-making process of the network. Consequently, our approach offers better insights into the higher-level causal mechanisms that shape the network's behavior, and allows us to provide a more abstracted, structured, and interpretable view of the causal dynamics intrinsic to the DNN's operations.

### 3.3 COUNTERFACTUAL GENERATION

Counterfactuals are hypothetical data instances that, if observed, would lead the model to provide a distinct decision. Crafting such instances is challenging due to the constraint that all counterfactuals should be realistic. Therefore, using generative models, specifically Generative Adversarial Networks (GANs) (Goodfellow et al., 2020), we aim to achieve this task by including such constraints into our training process. Given an input $x \in \mathbb{R}^d$ and a target model output $y^*$, we formalize the GAN-based counterfactual generation as:

**Generator ($G$):**

1. *Encoding the Input:* An encoder function $E_x$ is employed to map the input $x$ to a condensed latent representation $z_x = E_x(x)$.

2. *One-hot Encoding the Target Output:* The desired model output $y^*$, typically an integer label, undergoes a one-hot encoding transformation to produce a vector $o(y^*) \in \mathbb{R}^k$ where $k$ represents the number of classes:

$$o_i(y^*) = \begin{cases} 1, & \text{if } i = y^*; \\ 0, & \text{otherwise.} \end{cases}$$

3. *Concatenation:* The latent representation $z_x$ and the one-hot encoded target label $o(y^*)$ are concatenated: $z = [z_x; o(y^*)]$.

4. *Decoding to Counterfactual:* A decoder $D$ then processes this augmented latent vector to yield a counterfactual instance $x^* = D(z)$.

**Discriminator ($\mathcal{D}$):** The discriminator's role is to evaluate the authenticity of the crafted counterfactual $x^*$. Specifically, $\mathcal{D}$ discerns between the original data samples and the samples generated by $G$, verifying the realism of $x^*$.

During training, we optimize the GAN using a dual objective. This dual objective aims to ensure the authenticity of the generated counterfactual and to minimize the distance between the counterfactual $x^*$ and its nearest neighboring instance $x_{\text{nn}}$ with target label $y^*$ in the training dataset. This objective can be seen as a combination of a conventional GAN loss and a proximity measure (e.g., Euclidean distance) between $x^*$ and $x_{\text{nn}}$: $\mathcal{L} = (1-\lambda)\,\mathcal{L}_{\text{GAN}} + \lambda\,d(x^*, x_{\text{nn}})$, where $d(\cdot, \cdot)$ is the proximity function and $\lambda$ is a balancing coefficient, ensuring that generated counterfactuals are not only perceptually valid but also closely resemble genuine instances leading to the desired model prediction.

Employing generative models for the counterfactual generation task, as opposed to directly using the closest neighboring samples as counterfactuals, offers several key advantages. First and foremost, relying on such neighboring samples, while they may indeed represent real data points, would require storing large dataset, potentially leading to memory constraints and data privacy concerns. This could be particularly problematic in applications where storage is expensive or limited. On the other hand, generative models afford us the flexibility to create novel, yet still realistic, samples, that might not even exist in the original dataset. This generative capability allows for a broader exploration of the feature space and provides insights into regions of the decision boundary that could otherwise be overlooked. Further, from a computational standpoint, generative models offer a significant advantage, as searching through a dataset to identify the closest neighbors, especially in high-dimensional spaces, can be computationally intensive and introduce latencies. In contrast, by using a pre-trained generative model, we can generate counterfactuals on-the-fly, without the added computational cost of dataset searches. Consequently, our approach for counterfactual generation offers a compelling blend of realism, data efficiency, and computational performance.

## 4 EXPERIMENTS

In this section, we evaluate our proposed explainability method, TRACER, emphasizing both its causal discovery facet and its counterfactual generation approach described earlier. Our experiments are performed using the well-known MNIST dataset (Deng, 2012), a standard in image classification tasks, offering a collection of handwritten digits ideal for scrutinizing of our methodology. We use a pre-trained AlexNet (Krizhevsky et al., 2012) architecture as our MNIST classifier, and design a GAN architecture tailored to our counterfactual generation task. This GAN, depicted in Figure 2, consists of a CNN-based Generator for creating plausible, class-conditional counterfactuals, coupled with a CNN-based Discriminator analyzing the authenticity of the generated images. The GAN's generator is designed as follows:

(1) the encoder uses four convolutional layers to transform the input into a latent space, then merged with class information via one-hot label embeddings; (2) the decoder uses transposed convolutional layers to construct the counterfactual input from the augmented latent representation produced by the encoder. This counterfactual generator is trained using the Adam optimizer (Kingma & Ba, 2014) with a learning rate of $10^{-3}$.

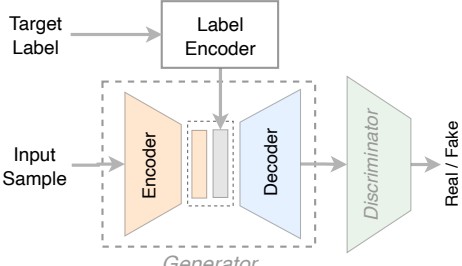

Figure 2: Counterfactual GAN architecture.

Through our experiments, we seek to provide a comprehensive understanding of TRACER's capabilities and the insights it offers into NN decision-making.

### 4.1 CAUSAL DISCOVERY

To evaluate the effectiveness of TRACER in uncovering the intricate causal pathways that govern decision-making in neural networks, the relationships between activations of different layers are

analyzed using their CKA similarities. Comparing activations produced by the original input and its corresponding interventions illuminates the effect of these interventions on neural network decisions. As showcased in Figure 3a, TRACER discerns layer groups forming causal nodes and identifies the causal links between them. For instance, eight activation outputs from the classifier are observed and analyzed for the AlexNet classifier, revealing inherent groupings based on similarity patterns across the network layers. This observation has led to the identification of four distinct causal nodes. Notably, the lack of causal connections between non-adjacent layer groups indicated a linear causal chain that informs the network's decision for the analyzed sample. Further visual insights from Figure 3b depict how individual features contribute to the network's final decision. For every causal node, we highlight the top contributing features (top convolution filter output or top-3 feature outputs for linear layers). Positive contributions are distinctly marked in blue, signifying features that positively influence the network's decision, while negative contributions are depicted in red, pointing out the features that negatively affect the decision.

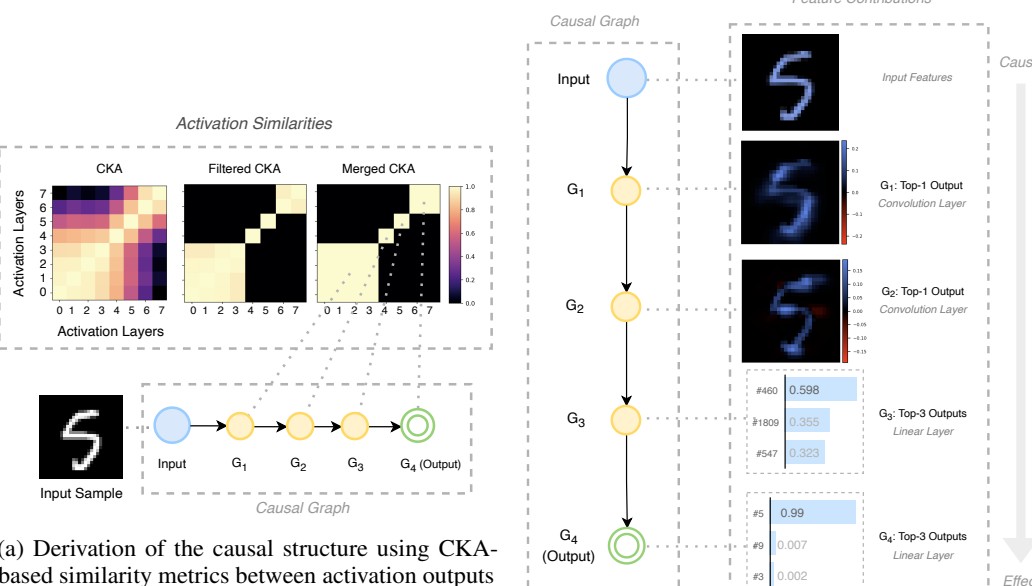

(a) Derivation of the causal structure using CKA-based similarity metrics between activation outputs from various layers. Nodes in the resulting causal graph symbolize layer groups, while the connections between them capture their causal relationships.

(b) Contribution of features within each causal node. Blue indicates positive contributions while red denotes negative contributions. The overlay on the input sample provides a cohesive visualization of how distinct features of the input affect the network's decision ("5" with 99% probability) via the causal mechanism discovered.

Figure 3: TRACER's causal analysis results for an MNIST sample classified by AlexNet.

## 4.2 COUNTERFACTUAL ANALYSIS

By performing counterfactual analysis, we aim to provide further insights into the decision-making processes of classifiers, especially during misclassifications. The generated counterfactuals, or alternative inputs, allow the identification of subtle features or patterns that influenced the model's decision. This enables us to determine the minimal changes needed to correct misclassifications. Through a comparison of the causal mechanisms uncovered for the misclassified sample with those for its counterfactuals, TRACER offers a deep understanding of the primary factors leading to the initial misclassification and the decision pathways that would result in the correct outcome. Such analyses can hint at limitations in a model's learned parameters, suggesting potential strategies for model improvement, such as refining the training set or implementing regularization techniques. In essence, counterfactuals offer both an intuitive understanding of model decisions and actionable insights for model enhancement. Appendix A presents a detailed counterfactual analysis for a misclassified MNIST sample using TRACER.

### 4.3 GENERALIZATION AND SCALABILITY

In this experiment, we highlight the broad adaptability of our approach across various neural network architectures and datasets. To this end, we expand our evaluations of TRACER to include an additional image recognition task, as well as a Network Intrusion Detection problem, explaining the decisions of both elementary NN architectures and complex structures such as ResNet-50 (He et al., 2016).

Given the wide variety and realisic nature of the samples in the ImageNet dataset (Deng et al., 2009), its classification results with the ResNet-50 architecture provide a solid benchmark for highlighting the limitations of existing explainability methods and comparing their performances to that of TRACER. For this comparison, we selected LIME (Ribeiro et al., 2016), SHAP (Lundberg & Lee, 2017), LRP (Bach et al., 2015), and Grad-CAM (Selvaraju et al., 2017), since they are among the most widely adopted and representative explainability methods in the literature. The results, depicted in Figure 4 show that while existing methods struggle to produce consistent explanations, TRACER provides coherent and comprehensive explanations that highlight the most important features and patterns that drive the classification decisions. Further comparison of these methods, discussed in Appendix B.1, highlight more distinctions between TRACER and existing methods, especially with DNN architectures exhibiting complex interactions.

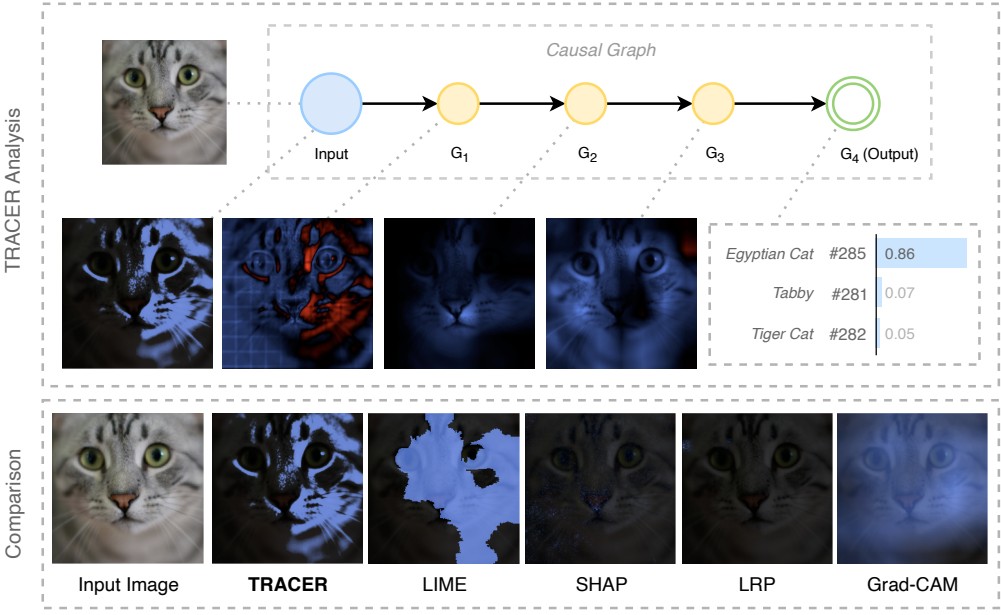

Figure 4: TRACER vs existing XAI methods using an ImageNet sample classified by ResNet-50. The second row shows feature contributions from different causal nodes, while the bottom row compares the explanations provided by different methods. Due to the sparsity of the results produced by SHAP and LRP, their explanations may require high-resolution color screens for proper visualization.

Diving deeper into the versatility spectrum, we challenge TRACER with the intricacies of structured data using the CIC-IDS 2017 dataset (Sharafaldin et al., 2018). This dataset, reflecting authentic network dynamics, unfolds a distinct set of challenges useful for evaluating explainability methods (e.g., diverse data types and intertwined correlations). In one notable instance, where a DDoS-attack-induced traffic is erroneously classified as benign (see Appendix B.2), TRACER identifies and elucidates features emblematic of the attack through its causal analysis. Specifically, TRACER reveals that features such as port numbers and data transfer dynamics are essential for the detection of such threats. Overall, the granularity and transparency of explanations provided by TRACER, especially in domains such as cybersecurity, accentuate its potential to build trust in critical applications.

### 4.4 BEYOND LOCAL EXPLAINABILITY

To evaluate TRACER's capacity for global explainability, we integrated individual local explanations to form a comprehensive view of a model's decision logic. For this task, we focused on a random

subset of the MNIST dataset, processed through the AlexNet architecture, to derive causal insights underpinning the classifier's decisions for all class samples. The results of this analysis, detailed in Appendix C, reveal significant redundancies within the classifier's architecture, allowing us to design compressed representations of the model to optimize the computational efficiency.

Table 1: Comparison of TRACER-assisted compressed models. $\theta$ represents the number of parameters of the models (in millions), and Speed indicates the inference time per sample (in milliseconds).

| Model | $\theta$ (M) | Size (MB) | FLOPs (M) | Speed (ms) | Accuracy (%) |
|---|---|---|---|---|---|
| AlexNet | 11.7 | 46.8 | 46.3 | $4.23^{\pm 0.4}$ | **99.64** |
| Compressed — C3 | 11.5 | 46.3 | 25.0 | $3.21^{\pm 0.3}$ | **99.64** |
| Compressed — C2 | 9.4 | 37.9 | 22.9 | $2.65^{\pm 0.1}$ | 99.57 |
| Compressed — C1 | **0.06** | **0.27** | **13.5** | $\mathbf{1.08^{\pm 0.1}}$ | 99.48 |

The characteristics and comparisons of these compressed models, reported in Table 1, show that the most refined model obtained exhibits a staggering 99.42% reduction in model size with only a 0.16% drop in accuracy. This highlights TRACER's potential for catalyzing practical innovations in DNN design and optimization, without undermining the predictive performance of these models.

## 5 DISCUSSIONS

In this study, we focused our evaluations of TRACER on white-box neural networks. However, its flexibility and design extend beyond, making it equally applicable to black-box models where the internal dynamics remain obscured and only the inputs and outputs are accessible. Under such constraints, TRACER remains valuable, offering two distinct avenues of exploration. First, it can analyze and quantify the influence of input features on the model's prediction. Alternatively, by using a surrogate white-box model, we can effectively approximate the underlying causal mechanisms driving the predictions. This adaptability underscores TRACER's potential in diverse environments.

The adaptability of TRACER is further highlighted by its compatibility with probabilistic models, which are known to be particularly apt at capturing the inherent uncertainties and randomness of certain real-world scenarios. To identify such layers with embedded randomness, TRACER runs the layers twice for the same input and checks for any variations in their outputs. When our approach is applied to these models, these detected exogenous variables are seamlessly included in the discovered causal mechanisms. This ensures a comprehensive understanding of the causal dynamics, especially in scenarios where randomness is intrinsic.

While the TRACER approach is highly parallelizable by design, its depth of analysis can introduce a balance between granularity — the precision of the causal analysis determined by the number of interventions generated for each sample — and computational efficiency. Delving deeper into interventions offers better causal insights but at the cost of higher computational requirements. This trade-off should therefore be adjusted depending on whether the emphasis is on detailed causal explanations or more overarching insights within constrained computational budgets.

## 6 CONCLUSION

In this paper, we introduced TRACER, a novel approach for illuminating the causal dynamics embedded within deep neural networks. Through seamless integration of causal discovery and counterfactual analysis, our methodology enables a deep understanding of the decision-making processes of DNNs. Our empirical results demonstrate TRACER's ability to not only identify the causal nodes and links underpinning a model's decisions, but also leverage counterfactuals to pinpoint the nuances that drive misclassifications, offering clear and actionable insights for model refinement and robustness. Beyond local explanations, we showcased the potential of our approach to capture the global dynamics of neural networks, leading to practical advantages such as novel and effective model compression strategies. Through our foundational principles and findings, we have ascertained that by producing intuitive, human-interpretable explanations, TRACER offers outstanding transparency to neural networks, significantly enhancing their trustworthiness for critical applications.

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

## A    DETAILED RESULTS: COUNTERFACTUAL ANALYSIS

The objective of counterfactual generation in the context of our research is to offer interpretable insights into the decision-making of deep neural networks, particularly in cases of misclassification. By examining the contrast between the original input and the generated counterfactual, we can potentially uncover subtle features or patterns that influence the model's decision, thereby pinpointing what changes might rectify misclassifications.

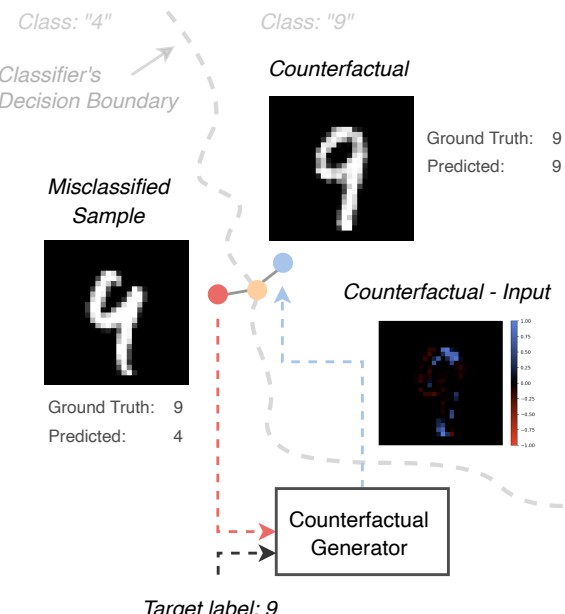

Figure 5: Illustration of a misclassified MNIST sample and its generated counterfactual.

As illustrated in Figure 5, given an initially misclassified input and a desired target label, our GAN-based counterfactual generator produces an alternative version of the input, which, when fed to the

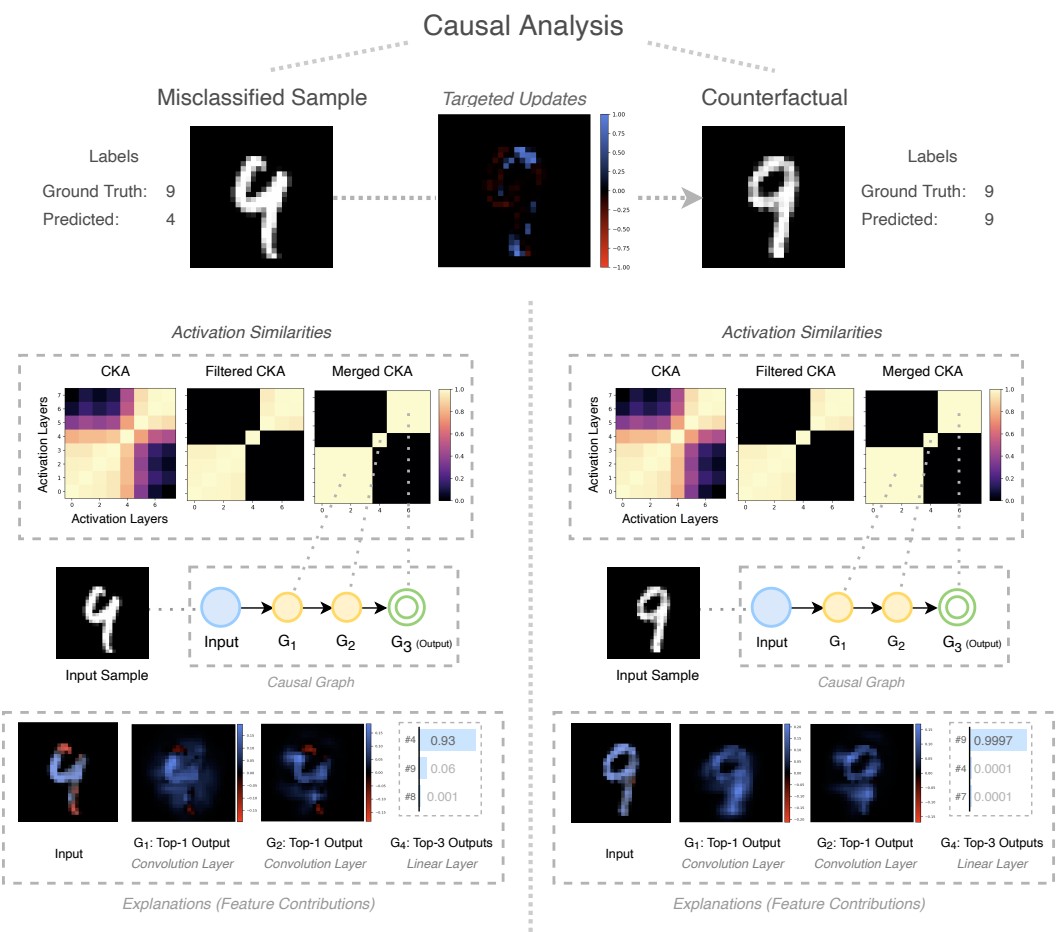

Figure 6: Comparison of an original misclassified input, the generated counterfactual, and the associated causal mechanisms. The variations between the original and counterfactual inputs highlight the pertinent features influencing the model's decision-making process. (Blue: Positive contributions; Red: Negative contributions; $G_i$: $i$-th Layer Group)

model, results in the desired outcome. The differences between the input and its counterfactual reveal the minimal modifications required for the classifier to produce the correct (desired) decision.

Through a side-by-side analysis of the causal mechanisms obtained from the predictions of the misclassified sample and its counterfactual, TRACER provides clear and profound insights. It not only reveals the primary contributors to the initial misclassification but also highlights via the counterfactual's analysis, the optimal neural pathways for the network to yield the correct (and desired) outcome. This detailed causal analysis is visually represented in Figure 6. Upon examination, we discern that a predominant portion of the input features, accentuated in blue, activate neurons that steer the classifier towards the produced outcome in both cases. However, the misclassified sample's causal analysis unveils a notably more extensive set of features that oppose the predicted outcome when contrasted with the counterfactual. This observation makes it evident that TRACER not only demystifies which parts of the input features support the misclassification (in blue) but also which features contradict this decision (in red). Intriguingly, while the causal graphs remain consistent for both inputs, the classifier's activations manifest pronounced differences. This insight suggests that the model's learned parameters might lack the flexibility to generalize enough to correctly discern the true label of the misclassified sample. To address this, potential avenues might include incorporating such misclassified instances into the training set or fine-tuning the model with regularization techniques to enhance its generalization capabilities.

This causal analysis reveals that our counterfactual generation method serves two main purposes. First, it provides an intuitive visualization for understanding the nuances of model decisions. Secondly, from a model development and refinement perspective, these counterfactuals can highlight potential vulnerabilities or biases in the model, guiding further training or fine-tuning endeavours.

## B    DETAILED RESULTS: GENERALIZATION

### B.1    IMAGE DATASETS

Here, we address the question of scalability of TRACER to large-scale image datasets. Given the challenges associated with the explainability of real-world images (e.g., the intricacies of pixel-level interactions, variances in image quality, or scale), we use for this task the MNIST and ImageNet (Deng et al., 2009) datasets, classified with the AlexNet and ResNet-50 architectures respectively. Using the ImageNet dataset, known for its vastness, diversity, and complexity, we show that TRACER overcomes the limitations of existing explainability methods. The explanations produced by TRACER and benchmark explainability methods are depicted in Figure 7, showing that while existing methods struggle to produce coherent and comprehensive explanations, TRACER consistently reveals the core features and patterns crucial for classification decisions. The effectiveness of our proposed approach becomes even more apparent when used with complex models like ResNet-50, as it still maintains its precision despite the intricate patterns leveraged by very deep networks, emphasizing its capability to elucidate the nuances of complex interactions within deeper architectures.

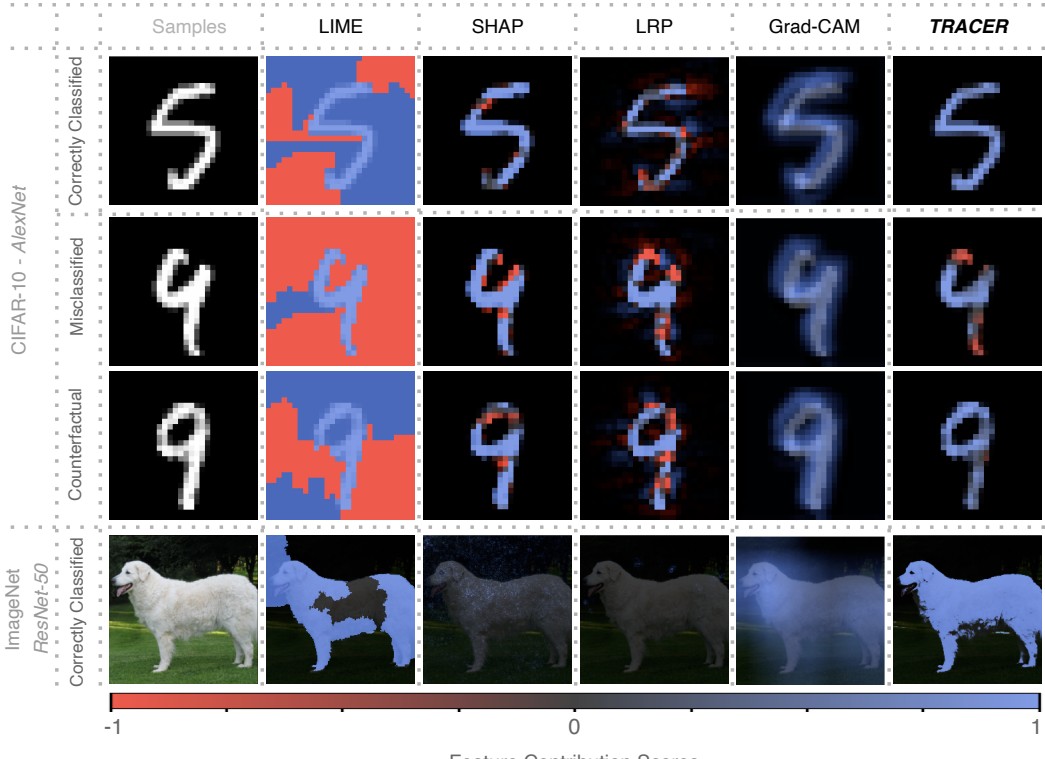

Figure 7: Comparison of TRACER results against existing explainability methods.

In contrast to TRACER,

- Every execution of *LIME* produces different explainability results due to its inherent stochastic nature, which hinders interpretability.
- *SHAP* and *LRP* explanations produce misleading results due to their sensitivity to model and dataset complexities, resulting in overly detailed or sparse attributions that do not always intuitively align with the underlying data patterns.

- As *Grad-CAM* explanations are based on the coarse spatial resolution of the final convolutional layer of a DNN, this method often leads to highlighting broader regions rather than precise feature-level contributions to the decision-making process.

- *LRP* and *Grad-CAM*, inherently designed for white-box DNNs, where internal model structures are accessible, face significant restrictions in terms of applicability and utility in scenarios involving black-box or proprietary models.

### B.2 TABULAR DATASETS

Transitioning from the realm of images, we further explored the efficacy of TRACER in the context of structured (or tabular) data. For this endeavour, we selected the CIC-IDS 2017 (Sharafaldin et al., 2018) network traffic dataset, which is representative of real-world network behaviors and patterns. This dataset poses its own set of challenges, distinct from image datasets, such as the mix of categorical and numerical attributes, the potential correlations between features, and the variance in feature scales.

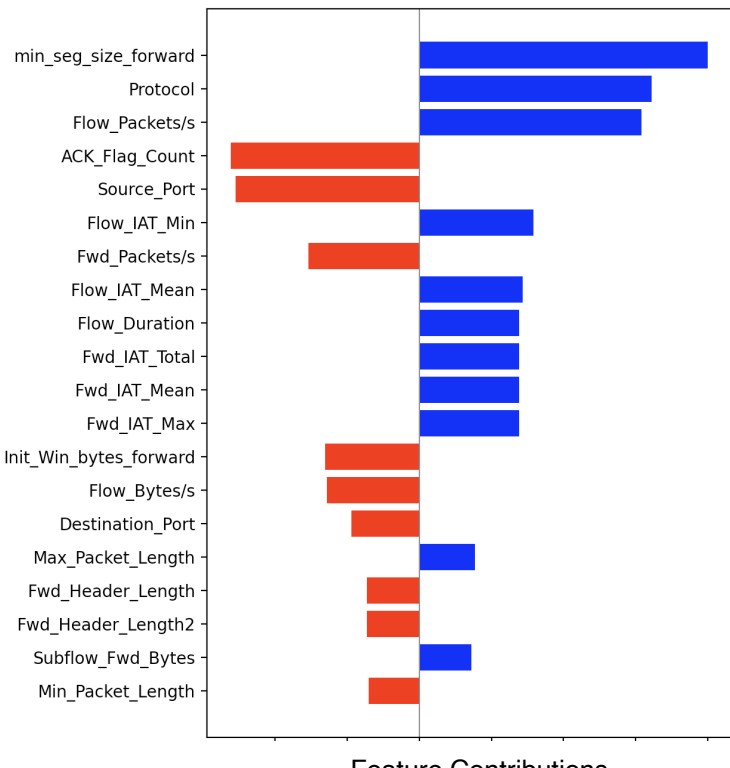

Figure 8: Explainability of tabular datasets with TRACER. A sample from a Network Intrusion Detection dataset is misclassified as benign traffic rather than its correct class (DDoS attack). Negative contributions are shown in red and positive contributions in blue for the top-20 features.

The results, presented in Figure 8 show exceptional coherence in TRACER's explanations. For the sample explained in this figure, where a network traffic generated during a DDoS attack is considered as benign traffic by a multi-layer feed-forward neural network classifier, we observe that the features indicative of an attack negatively contribute to the decision of the classifier. Specifically, the explanations provided tell us which features were found relevant for classifying this network traffic as an attack (i.e., Source/Destination Port numbers, frequency of communication, sizes of transferred data, etc.).

The richness and clarity of the causal explanations obtained by TRACER for such tasks make it particularly suitable given the criticality of network intrusion detection systems in ensuring cybersecurity,

where the ability to transparently understand and trust decisions can be indispensable for the practical viability of such systems.

## C  DETAILED RESULTS: GLOBAL EXPLAINABILITY

Given the effectiveness of TRACER in explaining neural network decisions for individual samples, we endeavour to evaluate its potential as a global explainability tool to paint a holistic picture of the model's decision-making. To this end, rather than solely relying on global explanations, which might overlook individual nuances, we adopt an approach that aggregates local explanations to derive a global perspective. Specifically, using TRACER, we perform local explanations on a strategically selected subset of the dataset, aiming to capture a representative understanding of the overall characteristics. For this experiment, we selected the MNIST dataset classified using the AlexNet architecture as before. While without loss of generality simply performing random sampling within all classes suffices for this experiment, by using different clustering algorithms (Settles, 2009; Olvera-López et al., 2010) or Proximally-Connected graphs (Diallo & Patras, 2023), more optimal sampling policies can also be adopted to identify and select the most influential samples. Our findings for this experiment revealed several remarkable facts about the potential of TRACER as well as using AlexNet for the MNIST classification task.

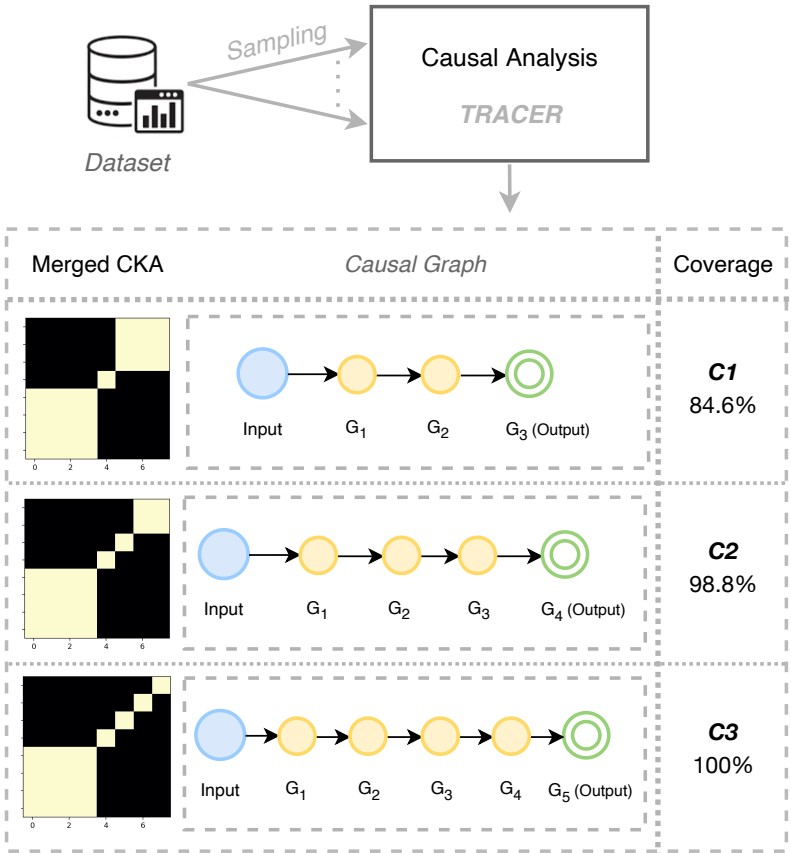

Figure 9: Global explainability with TRACER– Generalization of causal mechanisms across samples. The Coverage column indicates the percentage of analyzed samples that can be explained by distinct causal mechanisms.

Specifically, as shown in Figure 9:

1. About 85% of the samples could be concisely explained by a causal mechanism entailing merely 2 intermediate causal nodes. This level of generalization was unanticipated and showcases the simplicity underlying the model.

2. With just one additional causal node, the causal mechanism explains 99% of the classifications, bringing the total to 3 intermediate causal nodes.

3. To attain a full coverage, explaining 100% of the classifications, the complexity increases only marginally, requiring 4 intermediate causal nodes.

Encouraged by these insights into the causal underpinnings of AlexNet's decisions on the MNIST dataset, we ventured to create compressed representations of the original model. The objective was twofold: preserving the original model's accuracy while substantially reducing its computational complexity. Leveraging the knowledge distilled from TRACER, we crafted the corresponding compressed models and trained them on the identical training set as the original model (compressed models C1, C2, and C3, respectively corresponding to initial coverages of C1: 84.6%, C2: 98.8%, and C3: 100%). The results, presented in Table 1, show that the most compressed model achieved a staggering 99.42% reduction in model size, while only sacrificing a negligible 0.16% in accuracy, making it significantly more lightweight and computationally efficient.

By decoding the fundamental causal interactions within neural networks, this experiment shows that TRACER's capacity to provide global explanations and insights can also inspire practical applications such as model compression, without compromising the integrity of the predictions. Furthermore, it is worth noting that the compressed models derived through our approach remain fully compatible with existing and well-established compression methods such as quantization and pruning, further extending their efficiency and applicability across diverse deployment scenarios.

