# OpenReview forum: "Neural Networks Decoded: Targeted and Robust Analysis of Neural Network Decisions via Causal Explanations and Reasoning"
_ICLR.cc/2024/Conference — Submitted to ICLR 2024_

### Official Review · Reviewer_Gfne · 2023-10-31

**Soundness:** 2 fair
**Presentation:** 2 fair
**Contribution:** 2 fair
**Rating:** 3
**Confidence:** 5

**Summary:**

This paper is addressing the challenge of DNNs interpretability. The authors introduce TRACER, a novel explainability method
grounded in causal inference theory. They also introduce an approach to generate counterfactual explanations.

**Strengths:**

+ Definitely an interesting research question
+ Identification of causal sources... but unclear how this could be claimed as causal

**Weaknesses:**

- Baseline is reduced to SHAP, LIME, LRP and Grad-CAM. I would have expected more recent approaches as comparison
- Some comparisons should be done with causal based approaches
- Unsure on how that could scale easily
- Identification of causal sources... but unclear how this could be claimed as causal
- No experimentation on counterfactual which is claimed as contribution

**Questions:**

N/A

---

### Official Review · Reviewer_5com · 2023-11-01

**Soundness:** 1 poor
**Presentation:** 1 poor
**Contribution:** 2 fair
**Rating:** 3
**Confidence:** 3

**Summary:**

The authors propose a "causal" approach for explaining the behavior of neural networks.  In essence, the strategy involves generating perturbed inputs and then aggregating layers based on the results of a centered kernel alignment step.  The resulting "layer groups" are used to obtain local explanations and other visualizations.  Their proposed approach, TRACER, is also able to produce counterfactuals using a GAN model.

**Strengths:**

**Clarity**: English is generally good, the text is quite readable.

**Weaknesses:**

**Clarity**:  Unfortunately, the description of the proposed algorithm is too high level.  No proper formalization is given, meaning that it is difficult for me to figure out exactly what steps are carried out.  Section 3 would enjoy a more formal rewrite, where each block of the pipeline is described in detail, along with the inputs it takes and the outputs it produces.

**Originality**:  I am aware of causal approaches to explainability, but I have trouble understanding how they connect to the one proposed here (as it doesn't look causal to me).  Counterfactual generation approaches based on GANs (and other deep generative models) also already exist.

**Quality**:  The proposed approach is not described in sufficient detail to really understand its quality (I apologize if this sounds too harsh).  It is also not clear why the authors claim it is "causal" or why it performs "causal induction".  From what I understand, first a sufficient number of inputs are drawn by masking (training? test?) inputs, very much like LIME does, and then kernel alignment is applied.  I don't know why this procedure should identify the "causal dynamics" of the model.  To the best of my understanding, this is simply not explained.  The factual and counterfactual components are also quite detached from each other -- they feel like essentially independent contributions.

I am also afraid I found the experiments not entirely convicing.  Tracer is compared against baseline explainers (LIME, SHAP, LRP, Grad-CAM), which all come in several gradiants (especially SHAP), on two models (AlexNet, ResNet50) and two data sets (MNIST, ImageNet).  The qualitative evaluation of explanation quality is not really useful (how should I know how close the produced explanations are to the "true causal" explanations of model behavior?  These are unknown to me!)  This covers Figures 4 and 7.  I am also not sure how the additional visualization help.  I suspect the chosen data sets are not ideal.  For instance, CUB-200 might be a better target (it is a bird classification data set where images can be classified based on properties of bird parts, like beak color) especially for the layer group visualization.

**Significance**:  As things stand, I cannot claim this paper will have a deep impact on explainable AI, because the key feature it is supposed to bring about -- namely, causality -- does not appear to play a major role in the algorithm (although I could be wrong; as I mentioned this link is not obvious to me) or in the results.  Moreover, there already exists a vast literature on factual and counterfactual examples, including algorithms based on causality (e.g. see the works by Karimi and colleagues on algorithmic recourse, as well as all the follow ups), and it is not clear why I should prefer Tracer to them.

Other issues
----

**Introduction**: Considering there is substantial existing work on causal explainable AI (research by Geiger & co. and "Neural Network Attributions: A Causal Perspective" from 2019, for instance), counterfactuals, and algorithmic recourse, I substantially disagree with the authors claim that their "approach brings about a paradigm shift in neural network explainability by expanding our understanding of not just what happened, but why it happened, and what could have happened under different conditions".  This claim should be replaced with something more factual and realistic.

**Section 3.2**: The authors write that "we actively change the values of the classifier’s internal representations", but in the next subsection the appear to intervent on the inputs, not the embeddings.  I find this inconsistent.  It is also not clear why using a baseline value rather than marginalizing over the "masked" variables, nor what baseline value is best (if any).

**Section 3.2.1**: I agree that replacing an input variable by a constant breaks the link to possible confounding factors.  However, I don't think that this does "effectively nullify or modify the influence of the selected features or patches".  [Q1] I'd appreciate a clarification.

**References**:  Is the reference to Geiger et al's 2022 paper intended?  Did the authors mean to instead reference their 2021 work on causal abstractions?

**Questions:**

I do not have any precise questions.  I would be grateful if the authors could comment on the weakness I highlighted.  I am willing to increase the score if it turns out it have misunderstood something essential about the paper.

---

### Official Review · Reviewer_XGi8 · 2023-11-04

**Soundness:** 3 good
**Presentation:** 2 fair
**Contribution:** 2 fair
**Rating:** 3
**Confidence:** 4

**Summary:**

This paper introduces TRACER, an explainability method that compares the activations of different layers in DNNs in order to group them adjacently, before applying feature importance techniques to selected outputs.

**Strengths:**

- The paper makes first steps towards representing DNNs as causal models
- The explanations yielded by the method are compelling from a human-visual perspective
- Other explainability methods are provided for visual comparison
- The authors achieve strong compression results (high compression, minimal accuracy drops) with their method, though it is unclear how explanations are amassed to compress the network.

**Weaknesses:**

- Throughout, the method is described in a very general sense, with a lot of claims made regarding its efficacy that are not validated experimentally. I need not list them all as it is hard to find a paragraph without one. Here are a few examples:
  - Abstract: "*existing solutions often offer inconsistent, or overly simplified explanations*". Where is the comparison of TRACER's consistency vs other methods? Where is the justification that the steps involved in TRACER do not oversimplify the resultant model?
  - On the topic of using GANs for counterfactuals rather than nearest neighbors: "*First and foremost, relying on such neighboring samples, while they may indeed represent real data points, would require storing large dataset, potentially leading to memory constraints and data privacy concerns*". I'm interested in how the authors trained the GAN without storing the dataset. I agree that GANs are more likely to generate useful counterfactuals, but the proximity aspect of GAN counterfactuals is hardly emphasized, whereas vague claims about data privacy are made as the main justification.
  - "*Consequently, our approach for counterfactual generation offers a compelling blend of realism, data efficiency, and computational performance*". The use of generative models, and even more specifically GANs, has been proposed several times in the literature for counterfactuals. If there is something special about your design then it needs to be made clearer in the text with respect to the goals of TRACER.
  - The entirety of section 4.2. Most of this section explains why counterfactual explanations are useful. There is no substance to back the claims that "*TRACER offers a deep understanding of the primary factors leading to the initial misclassification and the decision pathways that would result in the correct outcome*". Instead, the appendix just shows feature importances for the original, misclassified input, alongside the counterfactual (at three layers). It is not obvious how this offers a "*deep understanding*" of the complex network internals.
  - No concrete network inspections or clear causal dynamics are presented, despite: "*Rather than merely highlighting influential features or approximating decision boundaries, our TRACER approach seeks to unravel the causal dynamics that steer DNN decisions*". Causality appears to be used as a way of masking the fact that the DNN still remains mysteriously complex.
  - DDoS Attack Experiments (deferred to Appendix): "*The results, presented in Figure 8 show exceptional coherence in TRACER’s explanations*", while the Figure shows standard feature importance measures on a single sample. No comparisons to other methods or solid metrics for multiple samples provided. Just a standard feature importance plot described later with words like "*richness*" and "*clarity*".
  - Compression results are not compared to any baselines.
- To summarize, the paper talks a lot about what the method can do, without adequately explaining how, and without providing actual scientific experimentation. It tends to use vague LLM-style justifications to support claims.
- The TRACER method introduces many steps, including the use of GAN-style counterfactual generators, but the computational efficiency of the method (an important metric, given how many steps are involved) is not documented or compared to existing methods.
- Further to this, it's unclear how attributions are actually computed. There is no algorithm or figure demonstrating how the framework does this. The method appears to just use some form of counterfactual reasoning to derive feature importances.
- Causality is used to motivate the paper, though there's no indication of what causal graph is exactly discovered besides a simple regrouping of the DNN's layers (adjacent layers are joined). Since this structure remains highly uninterpretable, what benefit does the causal framing of this problem bring?
- No code is provided and several experiments are discussed without full implementation details. In terms of ML explainability, I wholeheartedly hear the complaint from the authors that current feature importance techniques are lacking, but I am left somewhat clueless of what the method actually brings besides applying a feature attribution function to individual nodes of a network (determined by layer groups and top activations). It seems to serve several broad purposes, though with each described in minimal detail.
- Overall, I would recommend that the authors reserve hyperbole for when it is necessary, and instead focus on using those important spaces in the text to better describe exactly what is going on with the TRACER method. There's a chance the method does some special things, but there's no way of knowing until the entire framework is described properly.

**Questions:**

Please see above.

---

### Meta-Review · Area_Chair_4odn · 2023-12-10

**Metareview:**

All reviewers believe that this papers is poorly written, not clearly differentiated from existing literature, and unfit for publication. The authors declined the opportunity to advocate for their work.

**Justification For Why Not Higher Score:**

Because there is  nothing in the reviews to justify increasing the score and there is no rebuttal.

**Justification For Why Not Lower Score:**

Because there are no lower scores available.

---

### Decision · Program_Chairs · 2024-01-16

Reject